# MAX inactivation is an early event in GIST development that regulates p16 and cell proliferation

Inga-Marie Schaefer[1], Yuexiang Wang[1,†], Cher-wei Liang[1,†], Nacef Bahri[1], Anna Quattrone[1,2], Leona Doyle[1], Adrian Mariño-Enríquez[1], Alexandra Lauria[1], Meijun Zhu[1], Maria Debiec-Rychter[2], Susanne Grunewald[3], Jaclyn F. Hechtman[4], Armelle Dufresne[1], Cristina R. Antonescu[4], Carol Beadling[5], Ewa T. Sicinska[6], Matt van de Rijn[7], George D. Demetri[8], Marc Ladanyi[4], Christopher L. Corless[5], Michael C. Heinrich[9], Chandrajit P. Raut[10], Sebastian Bauer[3] & Jonathan A. Fletcher[1]

KIT, PDGFRA, NF1 and SDH mutations are alternate initiating events, fostering hyperplasia in gastrointestinal stromal tumours (GISTs), and additional genetic alterations are required for progression to malignancy. The most frequent secondary alteration, demonstrated in ~70% of GISTs, is chromosome 14q deletion. Here we report hemizygous or homozygous inactivating mutations of the chromosome 14q MAX gene in 16 of 76 GISTs (21%). We find MAX mutations in 17% and 50% of sporadic and NF1-syndromic GISTs, respectively, and we find loss of MAX protein expression in 48% and 90% of sporadic and NF1-syndromic GISTs, respectively, and in three of eight micro-GISTs, which are early GISTs. MAX genomic inactivation is associated with p16 silencing in the absence of p16 coding sequence deletion and MAX induction restores p16 expression and inhibits GIST proliferation. Hence, MAX inactivation is a common event in GIST progression, fostering cell cycle activity in early GISTs.

[1] Department of Pathology, Brigham and Women's Hospital, Harvard Medical School, 20 Shattuck Street, Thorn 528, Boston, Massachusetts 02115, USA. [2] Department of Human Genetics, KU Leuven and University Hospitals Leuven, Herestraat 49, Box 602, B-3000 Leuven, Belgium. [3] Sarcoma Center, Western German Cancer Center, University of Duisburg-Essen Medical School, Hufelandstrasse 55, 45122 Essen, Germany. [4] Department of Pathology, Memorial Sloan-Kettering Cancer Center, 1275 York Avenue, New York, New York 10065, USA. [5] Department of Pathology, Knight Cancer Institute, Oregon Health and Science University, 3181 Southwest Sam Jackson Park Road, Portland, Oregon 97239-3098, USA. [6] Department of Oncologic Pathology, Dana-Farber Cancer Institute, Harvard Medical School, 450 Brookline Avenue, Boston, Massachusetts 02215, USA. [7] Department of Pathology, Stanford University Medical Center, 300 Pasteur Drive, Stanford, California 94305, USA. [8] Ludwig Center at Harvard, Harvard Medical School and Department of Medical Oncology, Dana-Farber Cancer Institute, 450 Brookline Avenue, Boston, Massachusetts 02215, USA. [9] Portland VA Health Care System, Knight Cancer Institute, Oregon Health and Science University, 3181 Southwest Sam Jackson Park Road, Portland, Oregon 97239-3098, USA. [10] Department of Surgery, Brigham and Women's Hospital, Harvard Medical School, Boston, 75 Francis Street, Boston, Massachusetts 02115, USA. † Present addresses: Changzheng Hospital Joint Center for Translational Medicine, Institutes for Translational Medicine (CAS-SMMU); Key Laboratory of Stem Cell Biology, Institute of Health Sciences, SIBS, Chinese Academy of Sciences, Shanghai JiaoTong University School of Medicine; Collaborative Innovation Center of Systems Biomedicine, 320 Yueyang Road, Shanghai 200025, China (Y.W.); Department and Graduate Institute of Pathology, National Taiwan University Hospital and National Taiwan University College of Medicine, 7 Zhong-Shan South Road, Taipei, Taiwan 10002 (C.-w.L.). Correspondence and requests for materials should be addressed to J.A.F. (email: jfletcher@partners.org).

Activating mutations of the receptor tyrosine kinases $KIT$[1] or $PDGFRA$[2] are initiating or early events in most gastrointestinal stromal tumours (GISTs) and indeed are present in micro-GISTs, which are asymptomatic subcentimetre GIST lesions found in one-third of the general population[3]. Genetic progression from micro-GIST to malignant GIST results from stepwise accumulation of deletions in chromosome arms 14q, 22q, 1p and 15q, together with cell cycle dysregulating events and dystrophin inactivation[4–8]. These highly recurrent chromosomal deletions implicate losses of yet-unidentified tumour suppressor mechanisms in GIST progression. Of these, the 14q deletions are most frequent, observed in 60–70% of GISTs (including neurofibromatosis type 1 (NF-1)-associated GIST) as an early event in genetic progression[4–6,9,10].

Here we show that 14q deletions target the $MAX$ transcriptional regulator gene in early GISTs of various molecular origins ($KIT$-mutant, $PDGFRA$-mutant or $NF1$-mutant). These $MAX$ genomic-inactivating mutations are driver events, enabling GIST progression by loss of MAX expression, and consequent p16 silencing and cell cycle dysregulation.

## Results

**Genomic studies.** Targeted sequencing of 812 cancer-associated genes demonstrated somatic homozygous inactivating $MAX$ mutations in three of ten GISTs (Supplementary Data 1 and Supplementary Fig. 1a). The ten GISTs in this discovery set had $KIT$ mutations (seven cases), $PDGFRA$ mutations (two cases) and $NF1$ mutation (one case) (Supplementary Data 1). Apart from $KIT$ and $PDGFRA$, $MAX$ was the only other gene with demonstrable recurrent mutations in this discovery set. $MAX$ evaluations by Ion AmpliSeq sequencing, Sanger sequencing (Supplementary Fig. 1b), quantitative PCR (Supplementary Fig. 1c) and single-nucleotide polymorphism (SNP) arrays (Supplementary Fig. 1d) were performed in the same 10 GISTs and in 66 additional GISTs (Supplementary Data 2). This total set of 76 GISTs was shown to have mutually exclusive mutations involving the $KIT$, $PDGFRA$, $NF1$ and $SDH$ genes in 52 (68%), 8 (11%), 11 (14%) and 2 (3%) cases, respectively (Supplementary Data 2). These assays demonstrated somatic hemizygous or homozygous $MAX$-inactivating mutations in 16 of the 76 GISTs (21%), including 8 mononucleotide mutations and 8 larger-scale intragenic deletions (Fig. 1 and Supplementary Data 2). Non-neoplastic companion DNAs were $MAX$ wild type for seven of eight GISTs with mononucleotide $MAX$ mutations, showing that these were somatic mutations, and the $MAX$ mutation allelic frequency in the remaining case (case 19) was 0.7, consistent with a hemizgyous or homozygous somatic mutation. The $MAX$ mononucleotide mutations were nonsense ($N = 3$), loss of start codon ($N = 1$), frame-shift ($N = 1$), splice site ($N = 2$) and 5′-untranslated region ($N = 1$). The two splice-site mutations (cases 7 and 59) destroyed invariant splicing motifs, creating inactive $MAX$ transcripts with loss of exon 3 (case 7) or retention of intron 4 (case 59), as confirmed by reverse transcriptase–PCR, and—for case 7—also confirmed by genome RNA sequencing (Supplementary Fig. 2). The 5′-untranslated region mutation (case 19) was predicted to be functionally relevant by the PROMO.3 tool, with predicted disruption of transcription factor binding sites in the $MAX$ promoter. Multiple anatomically distinct specimens were studied in five patients (pts) with $MAX$-mutant GISTs and all had identical mutations, as shown by comparisons of primary GIST and subsequent metastases in two pts, and by comparisons of multiple metastases (two–ten metastases analysed per pt) in three pts (Supplementary Data 2 and Supplementary Fig. 3).

Among the overall study group of 76 GISTs, the GIST primary site was known for 71 pts, whereas primary site could not be determined in the remaining 5 pts who presented with disseminated intra-abdominal disease. $MAX$ genomic mutations were more common in non-gastric than gastric GISTs ($P = 0.001$ for the 71 pts with known GIST primary sites, two-tailed Fisher's exact test) and this association remained significant when GISTs with $NF1$ mutations—which are generally of small bowel origin—were removed from consideration ($P = 0.004$ for 61 pts with $NF1$-wild type GISTs of known primary sites).

**Protein studies.** MAX was assessed in each of the 76 GISTs by immunoblotting ($N = 75$) and/or immunohistochemistry (IHC) ($N = 22$). MAX inactivation was demonstrated in 38 of 75 GISTs (51%) by immunoblotting (Fig. 2) and was associated with $MAX$ genomic mutation ($P < 0.0001$, two-tailed Fisher's exact test). Likewise, MAX inactivation, as demonstrated by IHC in 14 of 22 GISTs (63%) (Fig. 3), was associated with $MAX$ genomic inactivation ($P = 0.0062$). Loss of MAX expression, when detected in any GIST metastasis, was detected in all others from the same pt (Supplementary Data 2 and Supplementary Fig. 4). Forty per cent of the GISTs with loss of MAX expression were classified, according to well-established clinicopathological criteria[11], as 'low-risk' and 'intermediate-risk' cases (Fig. 4), which are stages of GIST development that precede transition to clinically aggressive 'high-risk' GIST. These findings show that $MAX$ inactivation can be an early event in GIST biological and clinical progression. Further, MAX inactivation was detected in three of eight micro-GISTs, each of which had 14q deletion (Supplementary Fig. 5), confirming MAX dysregulation as an early event in GIST progression (Supplementary Table 1). Hemizygous or homozygous inactivating $MAX$ mutations were demonstrated in sporadic GISTs (11 of $64 = 17\%$) and in syndromic GISTs in individuals with NF-1 (5 of $10 = 50\%$) (Fig. 4 and Supplementary Data 2). Likewise, loss of MAX protein expression was demonstrated in both sporadic (31 of $64 = 48\%$)

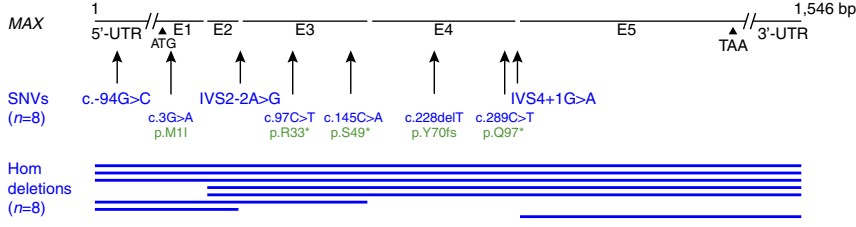

**Figure 1 | Genomic $MAX$ mutations in 76 GISTs.** Inactivating $MAX$ mutations were intragenic homozygous deletions (blue lines indicate deleted exons) and hemizygous mononucleotide alterations. Mutations are described according to international guidelines for sequence variant nomenclature by the Human Genome Variation Society (http://varnomen.hgvs.org). Annotations in blue are the nucleotide coding sequence mutations (indicated by 'c.'), whereas annotations in green are the resultant protein sequence mutations (indicated by 'p.'). All mutations affect both alternatively spliced forms of MAX, which encode 151 and 160 amino acid MAX isoforms.

and NF-1-associated GISTs (9 of 10 = 90%). GISTs occur in up to 25% of NF-1 pts, predominantly in the small intestine and often as multicentric tumours[12–14]. Our studies credential MAX

inactivation, as an early step in GIST progression, associated with *KIT* and *PDGFRA* gain-of-function mutations and *NF1* loss-of-function mutations.

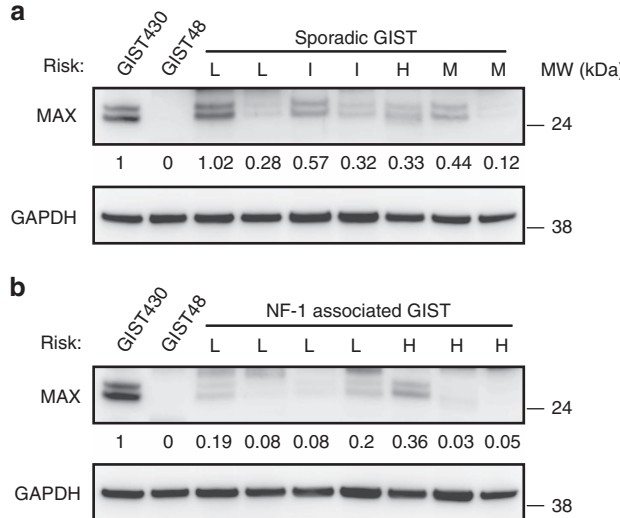

**Figure 2 | Loss of MAX expression in both sporadic and NF-1-associated syndromic GISTs.** MAX protein inactivation is demonstrated by immunoblotting GIST snap-frozen biopsies from sporadic (**a**) and NF-1-associated (**b**) cases. *MAX* wild-type GIST430 cell line and *MAX*-mutant GIST48 cell line are positive and negative controls, respectively. MAX inactivation was defined by expression level <0.4, normalized to GIST430. 'L' denotes low risk, 'I' intermediate risk, 'H' high risk and 'M' metastatic. The cases, from left to right in **a** are 21, 22, 32, 3, 39, 58 and 53, and in **b** are 73, 68, 71, 74, 76, 75 and 7. Residual MAX expression in *MAX*-mutant cases (53, 73, 68, 76 and 7) results from non-neoplastic cells (fibroblasts, endothelial cells and inflammatory cells) and from admixed precursor GIST cells that had not yet acquired the *MAX* mutations.

**Functional studies.** The GIST48 cell line has *MAX* inactivation due to homozygous deletion of *MAX* exons 1 and 2, and has loss of p16 (p16INK4A) expression. This cell line shows a localized *CDKN2A* deletion, which affects the p14ARF coding sequence (Supplementary Fig. 6) but lacks genomic alterations of the p16 coding sequence in *CDKN2A* and lacks *CDKN2A* methylation. *CDKN2A* ranked in the top 0.1% of genes differentially expressed after *MAX* restoration in GIST48 and was the highest ranking cancer-associated gene (Supplementary Data 3). In keeping with this evidence, p16 protein expression was strong and diffuse in GISTs lacking MAX or p16INK4A coding sequence mutations, but was undetectable in *MAX*-mutant GISTs, even in the absence of p16 coding sequence mutation (Supplementary Fig. 7). Inducible restoration of MAX expression in GIST48 upregulated *CDKN2A* transcript expression, restored p16 protein expression and inhibited RB1 phosphorylation (pRB1$^{\mathrm{Ser795}}$) (Fig. 5a) and cell proliferation (Fig. 5b,c). MAX restoration in GIST48 was not associated with significant enrichment of MYC-related expression signatures or with altered sensitivity to MYC:MAX inhibitor drugs (Supplementary Fig. 8), suggesting that MAX tumour suppressor roles in GIST are not necessarily MYC dependent.

## Discussion

All *MAX*-inactivating point mutations in this series were hemizygous, with allelic frequencies typically ∼0.67 in GISTs known to have loss of the other *MAX* allele and containing ∼20% non-neoplastic cells. Similarly, all *MAX* intragenic deletions were homozygous. These findings indicate that GIST progression is best served by complete loss of MAX function. Although the Ion AmpliSeq and HaloPlex assays detect ≥5% mutant alleles[15,16], we found no evidence for multiple

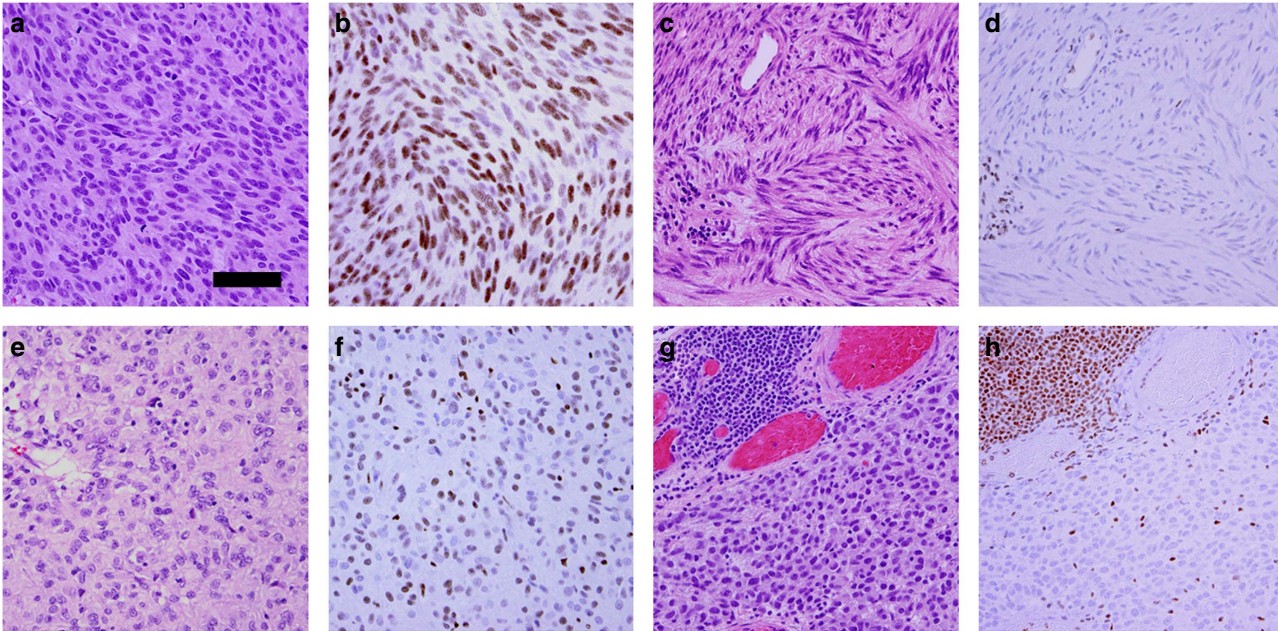

**Figure 3 | Loss of MAX protein expression can be present in cell subpopulations in early GISTs but is present in all cells from affected metastatic GIST.** Haematoxylin and eosin stains (**a,c,e,g**) and MAX IHC (**b,d,f,h**): case 67 with wild-type *MAX* (**a,b**) has retained MAX expression; case 29 (**c,d**), low-risk GIST with *MAX* mutation, has diffuse loss of MAX expression; case 3 (**e,f**), intermediate-risk GIST, has mosaic loss of MAX expression with admixed MAX-positive and MAX-negative cells; case 51 (**g,h**), metastatic GIST with *MAX* mutation, has diffuse loss of MAX expression. Positive internal controls for MAX expression in all cases are scattered inflammatory cells and fibrovascular cells, and—in case 51—a lymphoid aggregate at the upper left. Scale bar, 50 μm.

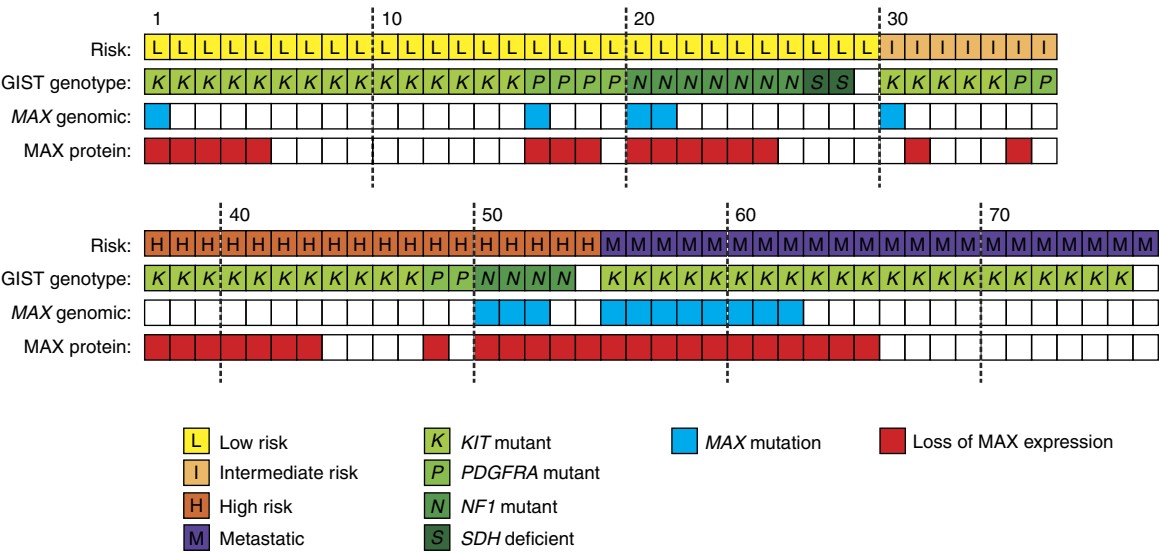

**Figure 4 | Summary of MAX genomic and protein aberrations in 76 GISTs.** Results are shown for both early and late stages of GIST progression (risk classifications), and for GISTs initiated by *KIT*, *PDGFRA*, *NF1* and *SDH* mutations.

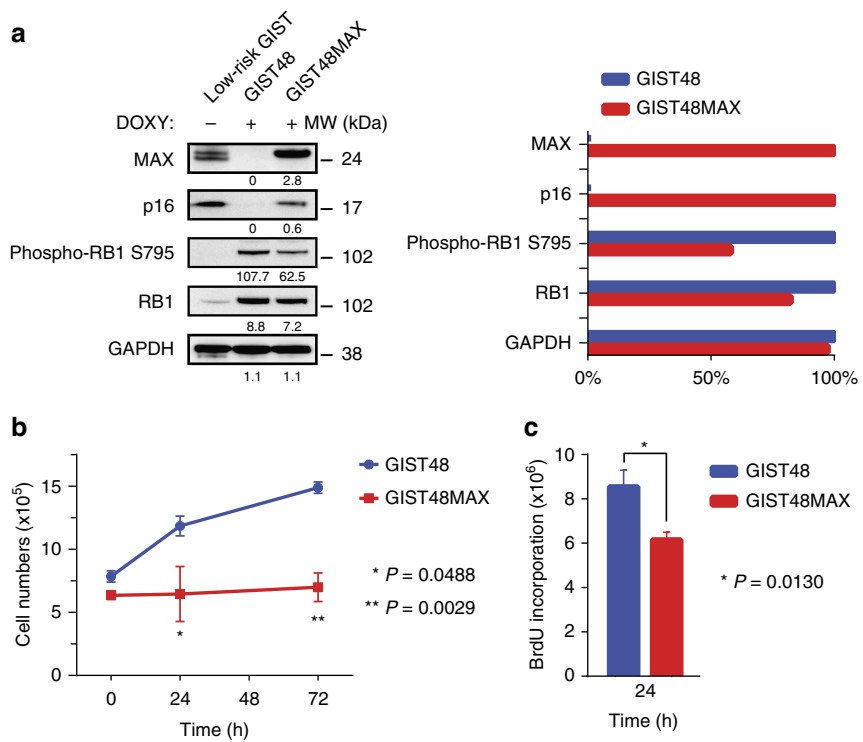

**Figure 5 | MAX restoration also restores p16 expression and function in GIST.** MAX restoration in *MAX*-mutant (homozygous deletion) GIST48 cell line restores p16 expression and inhibits CDK4/6-dependent phospho-RB1$^{Ser795}$; GAPDH serves as loading control and the bar graph normalizations of GIST48 versus GIST48MAX are with the higher value set to 100% (**a**). Restoration of MAX expression inhibits GIST48 growth, as assessed by cell counts at 24 h ($P = 0.0488$) and 72 h ($P = 0.0029$) (**b**), and inhibits GIST proliferation index, as assessed by BrdU incorporation at 24 h ($P = 0.0130$) (**c**). Tests were performed in triplicate. The error bars show s.d. (**b**) and s.e.m. (**c**).

*MAX*-mutant subclones within primary GISTs or between different metastases in a given pt. These observations indicate that a single *KIT/PDGFRA*-mutant/*MAX*-mutant or *NF1*-mutant/*MAX*-mutant subclone fosters biological and clinical progression in *MAX*-mutant GISTs. MAX is a helix-loop-helix leucine zipper transcription factor, which regulates cell proliferation, differentiation and apoptosis through heterodimerization with MYC-family proteins[17–19], but MAX homodimers also regulate transcription in a MYC-independent manner[20,21]. MAX has

tumour suppressor roles in a small subset of hereditary pheochromocytomas and in small cell lung cancer[21,22], and a recent report demonstrated *MAX* mutation in one *KIT/PDGFRA* wild-type, *NF1*-mutant, GIST[23]. Our study demonstrates that *MAX* mutations are common alterations in GIST, occurring at early stages of GIST development. We detected hemizygous and homozygous *MAX*-inactivating mutations in 21% of the 76 GISTs in this study, thereby demonstrating that GIST is the neoplasia with the highest known frequency of *MAX* tumour suppressor

mutations. Further, the true frequency of *MAX* mutations in GIST is likely to be >21%, given that homozygous deletions were a common mechanism of *MAX* inactivation. Homozygous deletions, particularly if small, will be difficult to detect in some early GISTs, where they are present in a subclone of the overall neoplastic proliferation.

We show MAX protein inactivation in ∼50% of GISTs, including micro and low-risk GISTs, which is additional evidence for MAX dysregulation as an early event in GIST development. MAX restoration in GIST48 cells inhibited GIST cell growth and upregulated *CDKN2A* expression, accompanied by p16 upregulation and inhibition of pRB1^Ser795. These findings suggest that *MAX* inactivation causes cell cycle dysregulation at an early point in GIST progression, probably enabling progression to GIST stages with greater proliferative potential. Altogether, our studies demonstrate frequent disruption of MAX tumour suppressive roles during early progression of *KIT*-mutant, *PDGFRA*-mutant and *NF1*-mutant GISTs.

## Methods

**Tumour and tissue samples.** Discarded, de-identified tumour specimens were obtained at Brigham and Women's Hospital and Memorial Sloan-Kettering Cancer Center, under protocols approved by the Dana-Farber/Brigham and Memorial Sloan-Kettering Cancer Center Institutional Review Boards. Informed written consent was obtained from all human participants.

**Cell lines.** GIST cell lines were established in the Fletcher laboratory and were validated against the initial biopsy material by molecular cytogenetics and sequencing verification of known unique gene mutations. Daudi cells were obtained from ATCC (Manassus, VA, USA). All cultures were shown to be mycoplasma free.

**Targeted sequencing.** Targeted sequencing was performed using the HaloPlex Target Enrichment System for Illumina sequencing (Agilent Technologies, Santa Clara, CA, USA) for a custom-designed set of 812 cancer-related genes. Data were analysed by SureCall software (version 2.0.7.0, Agilent Technologies) and Integrative Genomics Viewer (IGV) (version 2.3.25, Broad Institute). Targeted highly multiplexed PCR with semiconductor-based sequencing using the Ion AmpliSeq assay was performed as described previously[15], analysing the *MAX* gene (NM_002382) coding sequence, an additional two nucleotides adjacent to each exon, and 1 kb of upstream sequence. Amplicon size ranged from 125 to 275 bp (including primers) with an average of 243 bp. Inserts ranged in size from 77 to 230 bp (excluding primers), with an average of 194 bp. Ion AmpliSeq detection for homozygous deletions was performed after normalization to non-neoplastic DNA sequences and establishing cutoffs based on estimated presence of 30% non-neoplastic cells in low-/intermediate-risk GISTs and 20% non-neoplastic cells in high risk/metastatic GISTs. Deletion of at least nine consecutive amplicons and/or a ratio of <0.4 for markers located at either the 3′- or 5′-end of the gene in relation to all markers in a given case were defined as criteria for homozygous deletion.

**PCR analysis.** Genomic PCR and Sanger sequencing of *MAX* exons 1–5 (NM_002382) was performed using primers shown in Supplementary Table 2. Reverse transcription from tumour RNA was performed using the iScript cDNA Synthesis kit (Bio-Rad Laboratories, CA) and followed by PCR using Platinum PCR Super Mix (Life Technologies, Grand Island, NY, USA) with analysis by Sanger sequencing.

**Quantitative PCR.** Quantitative PCR of *MAX* exons 1, 3 and 4, and the flanking genes *FNTB* and *FUT8* was performed using 25 μl volume per reaction, containing 5 ng of genomic DNA, 100 nM forward primer, 100 nM reverse primer and 1 × iQ SYBR Green Supermix (Bio-Rad). The MyiQ single colour real-time detection system (Bio-Rad) was used for thermal cycling. Samples were run in triplicate with non-template control, human non-neoplastic cell DNA and GIST48 DNA with a known homozygous *MAX* deletion. Mixtures of non-neoplastic and *MAX*-mutant GIST48 DNA (30:70 and 20:80) were controls modelling detection of *MAX* homozygous deletions in human GIST biopsies (in which 10–30% of cells are nonneoplastic). Amplification accuracy was verified by melting curve analysis. The minimum threshold cycle (Ct value) generated by the MyiQ software (Bio-Rad) for each sample was used to calculate *MAX* copy number using Ct values for normal tissue and adjacent genes as reference.

**SNP arrays.** High-molecular-weight gDNA was isolated using QIAamp DNA Mini Kit (Qiagen, Valencia, CA, USA) and analysed by Affymetrix Cytoscan HD

2,600 K SNP array (Affymetrix, Santa Clara, CA, USA) with Affymetrix Chromosome Analysis Suite 2.0.

**Array comparative genomic hybridization.** Micro-GIST gDNAs were obtained from microdissected tissues and extracted using QIAamp DNA FFPE Tissue kits (Qiagen) following the manufacturer's protocol. Amplification was performed using a GenomePlex Tissue Whole Genome Amplification WGA5 kit (Sigma, Saint Louis, MO, USA). The post-WGA products were purified using a QIAquick PCR Purification Kit (Qiagen) and quantified using a NanoDrop ND-1000 Spectrophotometer. Comparative genomic hybridization was performed using a customized human array comparative genomic hybridization platform with 2 × 415 K coverage (Agilent Technologies). Four independent experiments were concurrently performed per template amplification and then mixed, to minimize amplification bias and allele dropout. Data were analysed using Agilent Technologies 10.5.1.1 Software.

**Protein blotting.** Whole-cell lysates were prepared as described previously[24] and protein concentrations were determined using the Bio-Rad protein assay (Bio-Rad). Electrophoresis, immunoblotting and chemiluminescence detection were as described previously[8]. Primary antibodies were directed against MAX (Santa Cruz Biotechnology, Dallas, TX, USA, C-17, 1:200 dilution), p16INK4A/CDKN2A (R&D Systems, Minneapolis, MN, USA, AF5779, 1:200 dilution), pRB1^Ser795 (Cell Signaling Technology, Danvers, MA, USA, 9301, 1:1,000 dilution) and GAPDH (Sigma, GAPDH-71.1, 1:5,000 dilution). Full-length blottings can be viewed in Supplementary Fig. 9.

**Immunohistochemistry.** Immunohistochemical staining for MAX was performed with the MAX C-17 antibody (Santa Cruz) at dilution of 1:1,500 on 4 μm thin sections prepared from formalin-fixed, paraffin-embedded tissue blocks after antigen retrieval using a citrate buffer pressure cooker protocol. Staining for p16 (p16INK4A) was performed using a mouse monoclonal antibody (dilution 1:2; clone E6H4, Ventana Medical Systems, Tucson, AZ, USA).

**MAX restoration.** Lentivirus preparations were produced by cotransfecting *MAX* construct (Broad Institute, clone ID BRDN0000560330, NM_002382.3) introduced into a destination vector pLXI_TRC401 (Alias pCW57.1 Dest, TRE-Gateway-No Tag) by LR Clonase (Thermo Fisher Scientific, Waltham, MA, USA) reaction and helper virus packaging plasmid pCMVΔR8.91 and pMD.G into 293T cells, as described previously[25]. Lentivirus was harvested at 24, 36, 48 and 60 h post transfection and virus titres were quantified and stored at −80 °C. GIST48 transductions were carried out overnight with polybrene 8 μg ml^−1 (Sigma) and transduced cells were selected for 9 weeks with puromycin (0.125 μg ml^−1), which was discontinued 7–10 days before analyses. MAX expression was induced in stably transduced cells with doxycycline (2.5 μg ml^−1) every 36 h.

**Gene expression profiling.** RNA sequencing was performed 24 h after MAX restoration in GIST48 cells using an Illumina HiSeq™ 2000 platform (Beijing Genomics Institute, Hong Kong). Parental GIST48 cells treated with doxycycline for 24 h served as control. Data analyses were with BGI-Tech Pipeline Version 3.1 and differentially expressed genes were identified using criteria false discovery rate ≤0.001 and abs(log2(MAX-/MAX+))≥1.

**BrdU uptake and CellTiter-Glo analyses.** Cells were plated in 96-well plates at 20,000 cells per well in growth medium and incubated overnight. Cells were treated with 2.5 μg ml^−1 doxycycline for 24 h (MAX-restoration) and parental GIST48 was the untreated comparator. For 5-bromodeoxyuridine (BrdU) incorporation proliferation analyses, BrdU was added to the cells for 24 h. BrdU incorporation, fixation and detection were performed using a BrdU Cell Proliferation ELISA as per the manufacturer's protocol (Roche Diagnostics, Indianapolis, IN, USA). BrdU incorporation was presented as % of untreated control. For CellTiter-Glo (Promega, Madison, WI, USA) viability analyses, MYC inhibitors versus dimethyl sulfoxide-only control were added for 3 days and ATP incorporation was then measured using a luminometer. All cell response assays were performed in triplicate wells, with the entire study replicated at least once.

**Cell counts.** Cells were trypsinized, resuspended in media and counted by haemocytometer.

**Statistical analyses.** Statistical analyses were performed using GraphPad Prism Software and Student's *t*-test and two-tailed Fisher's exact tests to compare two data sets.

**Data availability.** Haloplex targeted DNA sequencing data of the discovery cohort, Ion Ampliseq MAX sequencing data of all cases and gene expression data have been deposited in the Sequence Read Archive (accession number SRP096291). Cytoscan HD SNP array data have been deposited in the Gene Expression

Omnibus database (accession number GSE93077). All remaining data are available within the article and Supplementary Information files or available from the authors upon request.

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

## Acknowledgements

We thank Raymond DiDonato, PhD, from Agilent Technologies, for HaloPlex targeted cancer gene panel design and for other helpful advice with the HaloPlex analyses, and Jason Hornick, MD, PhD for assistance with the p16 IHC studies. This work was supported by grants from the US National Institutes of Health, including 1P50CA127003 (J.A.F. and G.D.D.), 1P50CA168512 (J.A.F. and A.M.-E.), the GIST Cancer Research Fund (J.A.F., M.C.H., C.L.C, C.R.A.), the Life Raft Group (J.A.F., M.v.d.R., M.C.H., C.L.C, S.B.), the Deutsche Krebshilfe (I.-M.S.), the VA Merit Review Grants 1I01BX000338-01 and 2I01BX000338-05 (M.C.H.) and the V Foundation Translational Research Grant (M.C.H.).

## Author contributions

J.A.F. supervised the research. J.A.F., I.-M.S., Y.W. and C.-w.L. conceived and designed the experiments. I.-M.S., Y.W., C.-w.L., N.B., A.Q., L.D., A.L., M.Z., S.G., J.F.H., C.B., M.L., C.L.C., M.C.H. and S.B. performed the experiments. J.A.F., A.M.-E., I.-M.S. and C.-w.L. performed statistical analysis. I.-M.S., Y.W., C.-w.L., N.B., A.Q., A.M.-E., S.G., J.F.H., A.D., C.B., M.v.d.R., M.L., M.C.H., S.B. and J.A.F. analysed the data. C.-w.L., C.R.A., M.v.d.R., M.L., E.T.S., C.P.R., S.B. and J.A.F. contributed reagents, materials and /or analysis tools. J.A.F. and I.-M.S. wrote the paper. M.D.-R., G.D.D. and C.L.C. provided scientific advice and helpful comments on the project. All authors read and approved the final manuscript.

## Additional information

**Competing financial interests:** C.L.C. has received consulting fees from ThermoFisher. The remaining authors declare no competing financial interests.

**Publisher's note**: 

