## [Peer Review File · Nature Communications]

Parts of this peer review file have been redacted as indicated to maintain the confidentiality of unpublished data.

Reviewers' comments:

Reviewer #1 (Remarks to the Author): Expert in Myc/Max

In this manuscript, the authors have shown that, depending on the subtype, ~17-50% of GISTs harbor a variety of hemizygous or homozygous point or deletion mutations involving the MAX gene. In most cases, this leads to a loss and/or significant reduction in Max protein. Max mutations and loss of expression appeared to be consistent in multiple tumors and metastases from the same patients thereby suggesting that Max mutation is an early event in tumor evolution. In the GIST48 cell lines, which lacks Max expression, also lack p16INK4a expression. The authors show that restoring Max expression leads to an up-regulation of p16INK4a and inhibited cell cycle progression

Major points

1. Re the apparent positive correlation between Max and p16INK4a. The CDKN2A locus also encodes p19ARF which is encoded by an alternate reading frame of the p16INK4a transcript. The authors should examine the status of p19ARF in their GIST samples to determine whether the effect of Max on the CDKN2A locus involves both proteins or is confined to p16INK4a.
2. What is the level of expression of p16 in tumors other than GIST48?
3. The loss of MAX would seem to imply that Myc expression would either be impaired or ineffectual. First, Myc levels should be examined. If possible, the authors should also compare several GIST cell lines both with and without MAX expression. In the former case, I would expect that the cell lines should up-regulate a Myc-responsive reporter and be sensitive to one or more of the well-accepted direct small molecule Myc inhibitors such as 10058-F4 or 10074-G5. In contrast, Max-negative cell lines should show none of these features unless they were forced to re-express Max.
This is actually an important point because, in most cell lines, Max expression far exceeds that of Myc such that hemizygous deletion should not have much bearing on Myc-regulated genes. It would therefore be interesting to determine the Myc response in GST48 cells following the restoration of Max expression.
4. Fig. 1. It is claimed in the text that this summarizes 16 Max inactivating mutations. The nomenclature however is confusing and should be changed. For example, what does C97>T p33R* mean?. Also, c.3G>A pM1l seems to suggest a mutation in codon 3? If so, then it should be recalled that Max encodes two isoforms, the longer of which encodes 9 amino acids between residues 2 and 3 in the shorter isoform (this is nicely seen in the Max immuno-blot profiles). Does this mutation affect only one of these isoforms?
5. Fig. 4 seems to indicate that some Max mutations allow for protein to be made. Is this the result of missense mutations or simply the expression of the non-mutated allele?
6. As best as I can determine, at least 3 of the mutations shown in Fig. 1 are indicated as occurring in either the 5'UTR region or in an intron. No discussion is provided as to how this leads to inactivation. Do the IVS mutations affect splicing? If so, evidence for this should be presented.
7. The first tumor in Fig. 2a, labeled "I" appears to express normal levels of Max relative to GIST430. How is this possible if the tumor contains a hemizygous inactivating mutation? Moreover, there is no way to relate the tumors being evaluated in Fig. 2 with those shown in Fig. 1.

Minor points

1. Line 43: define ICC
2. Fig. 2. The authors refer to a and b panels in the text but these are not labeled so in the actual figures. Also, there is no mention in the legend of what the notations L, I, M, H indicate. I assume they refer to Intermediate, high and low risk (Fig. 4) but this should be stated here.

Reviewer #2 (Remarks to the Author): Expert in GISTs

In this very succinct and well-documented paper the authors report MAX-gene loss-of-function mutations and MAX locus-involving losses resulting in biallelic inactivation in a tumor suppressor gene-like manner. MAX protein loss was shown by both immunoblotting and immunohistochemistry and was more common in NF1-associated than KIT-mutant GISTs. Loss of MAX protein lead into loss of CDKN2A expression in the absence of genomic alteration. Also, it was shown that MAX-restoration upregulated CDKN2A expression. It seems that MAX is at least one of the functionally significant tumor suppressor genes related to the long-known 14q loss in GIST and is a significant factor in GIST pathogenesis beyond KIT or NF1 alterations.

Reviewer #3 (Remarks to the Author): Expert in cancer genomics

In this manuscript, Schaefer et al performed targeted sequencing of 812 genes in 10 GIST samples. The authors reported recurrent mutations in MAX gene in 3/10 samples. In a validation set, they further further identified MAX mutations or homozygous deletions in 13/66 samples. Overall, losses of MAX expression were identified in 39/76 samples. Functional study suggested that MAX restoration in MAX-deleted cell lines inhibited GIST proliferation by upregulating p16 expression.

Major Comments

1. The criteria used to select MAX for validation studies should be described in greater detail. For example, what were some of the other recurrently mutated genes identified and is the frequency of MAX mutations significantly higher than the background mutation rate? How does the finding of MAX mutations compare to other results from published exome sequencing? It also would be useful to include the KIT and PDGFR mutation rates in this cohort (for both discovery and validation sets)
2. The authors should explicitly clarify if all the GIST sequencing in this study were from tumor only, or from tumor and matched normal genomic DNAs from the same patient. This reviewer was unable to get a clear sense of this from the paper.
3. Some of the MAX mutations in Figure 1 are suspect. What is the evidence that the -94 mutation in the 5' UTR is inactivating?
4. Page 3, line 85-87 "Twenty-one percent of the GISTs with loss of MAX expression were classified, according to well-established clinicopathological criteria, as "low-risk" and "intermediate-risk" GIST, ... (Fig 4)" The correct percentage seems to be 44% (16/36) as calculated from Fig 4.
5. Page 3, line 101-103 "The GIST48 cell line has MAX inactivation due to homozygous deletion of MAX exons 1 and 2, and has loss of p16 (p16INK4A) expression, even though the cells lack genomic alterations of the p16 coding sequence in CDKN2A and lack CDKN2A methylation." The same group had previously reported that GIST48 harbour a homozygous deletion in CDKN2A1. The conflicting explanations for the loss of p16 expression should be reconciled. The interpretation on the experimental studies should be toned down to reflect the findings in a single cell line.
6. Detailed clinical correlates of MAX mutations beyond low, intermediate, and high-risk categories would greatly strengthen the manuscript. For example, if GISTs with MAX mutations are more likely to progress to malignancy, or are associated with outcome, imatinib response/resistance, anatomical site etc.

7. Page 6, line 183-186, "Ion AmpliSeq detection for homozygous deletions was performed after normalization to nonneoplastic DNA sequences and establishing cutoffs based on estimated presence of 30% nonneoplastic cells in low/intermediate risk GISTs and 20% nonneoplastic cells in high risk/metastatic GISTs." How were the proportions of nonneoplastic cells estimated? Ideally they should be scored independently for each sample.

8. Since the authors have a wild-type MAX cell line, MAX silencing experiments (siRNA, shRNAs) should also be attempted.

9. In the RNA-seq experiments, are signatures related to Myc perturbed? This may help to clarify if the effects of MAX are Myc dependent or independent. This could be addressed by integrating the transcriptional data with Myc chip-seq binding data etc.

References

1 Eilers, G. et al. CDKN2A/p16 Loss Implicates CDK4 as a Therapeutic Target in Imatinib-Resistant Dermatofibrosarcoma Protuberans. *Molecular cancer therapeutics* 14, 1346-1353, doi:10.1158/1535-7163.MCT-14-0793 (2015).

The criticisms from the three expert reviewers were extremely useful, and – based on these criticisms – we have performed additional studies, including: 1) functional validations of *MAX* splice site and 5'-UTR mutations to confirm their inactivating nature; 2) cell proliferation assays to address responsiveness of *MAX*-inactivated GIST cell lines to *MYC* inhibition; and 3) further analysis of RNAseq data to assess *MYC* signatures. Importantly, we now emphasize (more clearly than before) that 15 of the 16 *MAX* genomic mutations demonstrated in these 76 GISTs were certain to be inactivating, because they were homozygous (or hemizygous) nonsense, frame-shift, start-codon mutation, and splice-site mutations, or large intragenic deletions. The remaining mutation is a *MAX* homozygous promoter mutation, which – as we now show – is predicted to be deleterious.

In responding to the Reviewers' critiques, we felt that several of the requisite new studies were particularly informative and should be represented by additional figures. To this end, we have added 4 new supplementary figures (Supplementary Figs. 2, 6, 7, and 8).

Our point-by-point responses are as follows:

REVIEWER 1:

Major points

1) *“Re the apparent positive correlation between Max and p16INK4a. The CDKN2A locus also encodes p19ARF which is encoded by an alternate reading frame of the p16INK4a transcript. The authors should examine the status of p19ARF in their GIST samples to determine whether the effect of Max on the CDKN2A locus involves both proteins or is confined to p16INK4a.”* We are grateful to Reviewer 1 for raising this key point. Very germane to this point, the inactivation of cell cycle checkpoints turns out to be a multistep process in GIST, and subsequent genomic alterations (after *MAX* inactivation) preclude a simple answer to this question in our GIST models. In the present work, we have shown *MAX* inactivation as an early step in GIST development (likely giving rise to slightly increased proliferation in a still-benign early GIST). However, GISTs require additional mutations to drive progression to a high-grade and clinically malignant phase. [redacted]. In the revised paper, we now include an example of this: a high resolution SNP analysis of GIST48 (new **Supplementary Fig. 6**) that acquired homozygous deletion of the p14ARF and p15INK4B ORFs, although not the p16INK4A ORF, subsequent to the *MAX* inactivation. These subsequent events make sense in the context of our demonstrated *MAX* roles (ie, that there is no added value in subsequent genomic mutation of p16INK4A). At the same time, these subsequent events, while fascinating to unravel (and the subject of a project we hope to complete to first stage in 2017) confound evaluations of *MAX* relevance for p14ARF expression in many GISTs. Such studies are further hampered because

available p14ARF antibodies (we have tested many) are not optimally specific in IHC or western blot studies.

(We have gone to some length in the reply, above, because there is also relevance to other excellent suggestions from the Reviewers).

2) “*What is the level of expression of p16 in tumors other than GIST48?*” For the reasons outlined above (response to point #1), depending on the *MAX* and *CDKN2A* genomic status, p16 levels are variable in advanced GISTs. GISTs that are wildtype for *MAX* and *CDKN2A* express p16 strongly, at levels seen for the low-risk GIST in Fig. 5a, whereas GISTs with genomic *CDKN2A/p16* or *MAX* inactivation lack p16 expression. To give a better sense for the p16 expression (diffuse and strong) in GISTs lacking *CDKN2A/p16* or *MAX* mutations, we have added a figure (new **Supplementary Fig. 7**) showing typical strong *MAX* and p16 expression in a GIST wildtype for *MAX* and *CDKN2A* vs. complete loss of p16 expression in a GIST with *MAX* mutation but lacking p16 coding sequence mutation or methylation.

3) “*The loss of MAX would seem to imply that Myc expression would either be impaired or ineffectual. First, Myc levels should be examined. If possible, the authors should also compare several GIST cell lines both with and without MAX expression. In the former case, I would expect that the cell lines should up-regulate a Myc-responsive reporter and be sensitive to one or more of the well-accepted direct small molecule Myc inhibitors such as 10058-F4 or 10074-G5. In contrast, Max-negative cell lines should show none of these features unless they were forced to re-express Max. This is actually an important point because, in most cell lines, Max expression far exceeds that of Myc such that hemizygous deletion should not have much bearing on Myc-regulated genes. It would therefore be interesting to determine the Myc response in GIST48 cells following the restoration of Max expression*”. To evaluate whether *MAX*-inactivated GIST cells might be less responsive to *MYC*-inhibition, we performed cell viability assays (CellTiter-Glo) in *MAX*-mutant GIST48 vs. *MAX*-wildtype GIST430, after treatment with *MYC*-inhibitors 10058-F4 and 10074-G5. The positive control was Daudi. *MYC*-inhibitor response was not significantly different in the *MAX*-negative vs. *MAX*-restored GIST48, whereas both these states were significantly less response than the Daudi control cells (new **Supplementary Fig. 8**). These interesting findings, in conjunction with the *MYC*-signature evaluations (see Reviewer 3, point #9, below) suggest that *MAX* tumor suppressor roles, in GIST, are not necessarily *MYC*-dependent.

4) “*Fig. 1. It is claimed in the text that this summarizes 16 Max inactivating mutations. The nomenclature however is confusing and should be changed. For example, what does C97>T p33R* mean? Also, c.3G>A pM1I seems to suggest a mutation in codon 3? If so, then it should be recalled that Max encodes two isoforms, the longer of which encodes 9 amino acids between residues 2 and 3 in the shorter isoform (this is nicely seen in the Max immuno-blot profiles). Does this mutation affect only one of these isoforms?*” These are descriptions of the coding sequence (genomic) alteration, which is denoted by “c.” and the protein sequence consequence, which is denoted by “p.”. These are as per international guidelines for sequence variant nomenclature by the Human Genome Variation Society (<http://varnomen.hgvs.org>), and we now state that in the Fig. 1 legend. For example, of the mutations mentioned by Reviewer 1, the first has a “c.97C>T” genomic mutation, which means the 97th nucleotide of the *MAX* coding sequence changes from C to T, resulting in “p.33R*”, ie the 33rd codon of the *MAX* protein changes from an arginine to a STOP. In the case of “c.3G>A” the 3rd nucleotide of the coding sequence changes from G to A, resulting in “p.M1I”, ie loss of the *MAX* methionine-encoding start codon. We have also now revised Fig. 1 by color-coding the mutation descriptions (nucleotide coding sequence mutations in blue, resultant protein alterations in green) and we state this in the revised figure legend, so as to make these annotations easier to follow. Also, we now state in the figure legend that all mutations affect both *MAX* isoforms (none of the mutations were restricted to the alternatively spliced *MAX* exon 2).

5) “Fig. 4 seems to indicate that some Max mutations allow for protein to be made. Is this the result of missense mutations or simply the expression of the non-mutated allele?” There were no missense MAX mutations in our GIST cohort. Rather, all of the intragenic mutations were of the completely inactivating type. The one case (Fig. 4) with genomic MAX inactivation (homozygous deletion of the entire coding sequence) but residual MAX protein expression is case 31 (Supplementary Table 2). As shown in Fig. 2, several of the MAX-mutant GISTs retained MAX expression when evaluated by immunoblotting of frozen tumor biopsies. The sources of MAX protein expression in these MAX-mutant cases are: 1) the nonneoplastic cells (fibroblasts, endothelial cells, and inflammatory cells) which comprise 20% of the cells in clinical GISTs (but a somewhat higher percent of the overall cells in lower-grade GISTs); and 2) GIST cells that represent early points in tumor progression, before the MAX mutation was acquired, and which are admixed with the MAX-mutant cells in a primary GIST. In case 31, it is likely that both these mechanisms contributed to classification of the case as “MAX-positive”, because it was an early GIST (intermediate risk). We have now revised the Fig. 2 legend, to explain why immunoblotting detects residual MAX protein expression (albeit, typically at very reduced levels) in primary GISTs with MAX mutations.

6) “As best as I can determine, at least 3 of the mutations shown in Fig. 1 are indicated as occurring in either the 5'UTR region or in an intron. No discussion is provided as to how this leads to inactivation. Do the IVS mutations affect splicing? If so, evidence for this should be presented”. Yes, both of the “IVS” (splice site) mutations destroyed normal splicing options. One of these replaced the “a” in the obligate “ag” at the end of an intron, and the other replaced the obligate “g” at the first nucleotide of an intron. We have now performed RT-PCR, which we summarize in the revised Results section, showing that IVS2-2A>G (case 7) resulted in loss of MAX exon 3, whereas IVS4+1G>A (case 59) resulted in retention of intron 4, creating nonfunctional MAX transcripts. To confirm this, we have also now performed RNAseq in case 7 (IVS2-2A>G), and we have added a Supplementary Figure (new **Supplementary Fig. 2**), showing the dramatic loss of MAX exon 3 transcript, resulting from the splicing aberration in that case.

We also now provide predictive evidence, using Promo.3 bioinformatics analysis, that the 5'-UTR mutation is expected to cause deleterious alterations of transcription factor binding sites in the MAX promoter. Further, this GIST (case 19) had substantial loss of MAX protein expression. In summary, the evidence is consistent with a causal role of the 5'-UTR mutation in MAX downregulation, and is definitive for a causal role in the other 7 cases with MAX mononucleotide mutations.

7) “The first tumor in Fig. 2a, labeled “I” appears to express normal levels of Max relative to GIST430. How is this possible if the tumor contains a hemizygous inactivating mutation? Moreover, there is no way to relate the tumors being evaluated in Fig. 2 with those shown in Fig. 1”. We have revised the Figure legend, now providing definitions for the abbreviations (“I” denotes Intermediate risk rather than Inactivating), and now specifying the cases, so that the MAX mutations can be cross-referenced in Supplementary Table 2. As an example, the first tumor is an “I” (intermediate risk) GIST, case 32, which is MAX-wildtype (Supplementary Table 2) and retains MAX expression.

Minor points

1) “Line 43: define ICC”. We now define ICC as the abbreviation for interstitial cells of Cajal (page 2). ICC are the pacemaker cells that coordinate orderly smooth muscle contraction in the GI tract.

2) “Fig. 2. The authors refer to a and b panels in the text but these are not labeled so in the actual figures. Also, there is no mention in the legend of what the notations L, I, M, H indicate. I assume they refer to Intermediate, high and low risk (Fig. 4) but this should be stated here”. We have revised Fig. 2, and the Fig. 2 legend, accordingly.

REVIEWER 2:

(No revisions were requested).

REVIEWER 3:

Major Points

1) “The criteria used to select MAX for validation studies should be described in greater detail. For example, what were some of the other recurrently mutated genes identified and is the frequency of MAX mutations significantly higher than the background mutation rate? How does the finding of MAX mutations compare to other results from published exome sequencing? It also would be useful to include the KIT and PDGFRA mutation rates in this cohort (for both discovery and validation sets)”. In the discovery cohort of 10 GISTs, 7 tumors had oncogenic KIT mutations (cases 1, 4, 5, 6, 8, 9 and 10), 2 tumors had oncogenic PDGFRA mutations (cases 2 and 3), and the remaining case had NF1 loss-of-function mutations (case 7). Beyond the MAX mutations (3 cases) there were no other recurrent mutations detected. In the validation set of 66 GISTs, 45 had oncogenic KIT mutations (68%), 6 had oncogenic PDGFRA mutations (9%), 10 had inactivating NF1 mutations (15%), and 2 had SDH mutations (3%): these were mutually exclusive groups. [We say the KIT and PDGFRA mutations were oncogenic, because these mutations have been shown previously – by our group and others – to have gain-of-function auto-phosphorylation properties]. We now include these important contextual data in the manuscript text. GISTs have been reported to have relatively quiet genomic landscapes, with no demonstrated recurrent gene (mutation) targets, to date, other than the alternate “primary” initiating mutations (KIT, PDGFRA, NF1, SDH genes) and the cell cycle regulatory gene mutations (see response to Reviewer 1, point #1) that occur at relatively late steps of primary GIST progression. Therefore, the finding of highly recurrent MAX inactivating mutations is a novel and major advance in the GIST field. MAX aberrations have not been detected in previous GIST exome-sequencing studies; in part (presumably) because the number of study cases was not high, and because the studies included low-risk GISTs in which the allelic frequency of MAX mutations (present in only a subset of the tumor cells) might be low and escape detection cut-offs. Another consideration is that the published exome screening studies did not concurrently evaluate larger-size intragenic MAX inactivating deletions, as was done in our study. In sum, the frequency of MAX mutations in GIST is high, surpassing that of all other recurring mutational targets other than KIT.

One study has identified a single KIT/PDGFRA wild-type, NF1-mutant, GIST with MAX mutation (Belinsky M. G. *et al.* Somatic loss of function mutations in neurofibromin 1 and MYC associated factor X genes identified by exome-wide sequencing in a wild-type GIST case. *BMC. Cancer* **15**: 887, 2015). However, motivated by a meeting presentation of some of our MAX data, we are aware that two strong GIST research programs have done targeted MAX sequencing in GIST, and have confirmed our results, and are preparing those data for publication (as targeted smaller-scale genomics studies which do not include our functional insights into biologic relevance of MAX inactivation in GIST).

2) “The authors should explicitly clarify if all the GIST sequencing in this study were from tumor only, or from tumor and matched normal genomic DNAs from the same patient. This reviewer was unable to get a clear sense of this from the paper”. We now specify that matched normal DNAs were sequenced for 7 of the 8 patients with MAX mononucleotide genomic mutations, showing that each of these mutations was somatic (not germline). Further, we state that the MAX mutation allelic frequency (70%) for the remaining mononucleotide mutation is consistent with a homo/hemi-zygous mutation of somatic origin.

3) “Some of the MAX mutations in Figure 1 are suspect. What is the evidence that the -94 mutation in the 5’ UTR is inactivating?” This point is addressed in our response to Reviewer 1, Major Point #6, and we have revised the manuscript accordingly.

4) “Page 3, line 85-87 ‘Twenty-one percent of the GISTs with loss of MAX expression were classified, according to well-established clinicopathological criteria, as “low-risk” and “intermediate-risk” GIST, ... (Fig 4)’ The correct percentage seems to be 44% (16/36) as calculated from Fig 4”. Thank you for noticing this error! In total, actually 16 of 40 GISTs (including sporadic and NF1-associated GIST) with loss of MAX expression were classified as low or intermediate risk. We have corrected this. We have also revised Fig. 4 to show that 40 GISTs had loss of MAX expression by immunohistochemistry and/or immunoblotting.

5) “Page 3, line 101-103 “The GIST48 cell line has MAX inactivation due to homozygous deletion of MAX exons 1 and 2, and has loss of p16 (p16INK4A) expression, even though the cells lack genomic alterations of the p16 coding sequence in CDKN2A and lack CDKN2A methylation.” The same group had previously reported that GIST48 harbour a homozygous deletion in CDKN2A¹. The conflicting explanations for the loss of p16 expression should be reconciled. The interpretation on the experimental studies should be toned down to reflect the findings in a single cell line.” Please see our response to Reviewer 1, point #1, which addresses this. We have now added a Supplementary Figure (**Supplementary Figure 6**) showing that although GIST48 does not have homozygous deletion for the p16 coding sequence (as we had stated) it does have homozygous deletion involving the 5’ end of CDKN2A, thereby disrupting the p14ARF coding sequence. We have made corresponding clarifications, in the Abstract, specifying that when we speak to absence of CDKN2A deletion, this is specifically the p16 coding sequence. We also now emphasize (final paragraph of text) that these findings are based on analyses in a single cell line (GIST48).

6) “Detailed clinical correlates of MAX mutations beyond low, intermediate, and high-risk categories would greatly strengthen the manuscript. For example, if GISTs with MAX mutations are more likely to progress to malignancy, or are associated with outcome, imatinib response/resistance, anatomical site etc.” We have added a paragraph that underscores one clinical correlate, which is that MAX mutations are more common in non-gastric GISTs (gastric primaries are the most common subset of GISTs). This association was highly significant, even when NF1-mutant GISTs (which are invariably non-gastric) were removed from the analyses. Non-gastric GISTs – overall – are more aggressive clinically than gastric GISTs, suggesting that MAX mutations could correlate with worse prognosis. However, this possibility will need to be tested in future, larger, population-based studies. Our findings credential MAX inactivation as an early event in GIST development, so it is clear that MAX inactivation does not specifically identify an already-aggressive subset of GIST. Further, we would prefer not to send a message that MAX mutations are more common in advanced (highly-malignant) GISTs, because biologic (and technical) considerations might confound detection of MAX mutations in early GISTs, particularly where the mutations are newly acquired and present in only a small subset of the neoplastic cells.

7) “Page 6, line 183-186, ‘Ion AmpliSeq detection for homozygous deletions was performed after normalization to nonneoplastic DNA sequences and establishing cutoffs based on estimated presence of 30% nonneoplastic cells in low/intermediate risk GISTs and 20% nonneoplastic cells in high risk/metastatic GISTs.’ How were the proportions of nonneoplastic cells estimated? Ideally they should be scored independently for each sample”. The rate of 20-30% nonneoplastic DNA was based on H&E stains of FFPE sections of the study cases, and was supported by the allelic frequencies of the homozygous or heterozygous primary mutations (KIT, PDGFRA, NF1 or SDH genes) that were demonstrated in 97% of the cases. We did not include cases in which the apparent percentage of nonneoplastic cells exceeded 30%. The tumor DNAs were from the same specimens that underwent histologic review, but we did not attempt to refine the nonneoplastic cell estimates beyond 20% (high-risk/metastatic cases) or 30% (low/intermediate risk cases), because precisely estimating the nonneoplastic cells at the single-digit level is not possible, and because the DNAs were isolated from frozen material in the same surgeries (not the exact FFPE sections used for histologic review), such that even higher-resolution estimates of % nonneoplastic cells in the DNA studies could be misleading, due to intratumoral heterogeneity.

8) “Since the authors have a wild-type MAX cell line, MAX silencing experiments (siRNA, shRNAs) should also be attempted”. We attempted MAX silencing in MAX-wildtype GIST cell lines, using a panel of shRNAs. However, these studies were uninformative because all of the MAX shRNAs had consequential off-target effects, as evidenced by growth perturbations in MAX-negative controls. An equally important confounding factor in these studies was our realization that the relevant GIST lines had cell cycle regulatory mutations (see discussion in response to Review 1, point #1) acquired at a later stage in GIST progression, that render MAX manipulations largely irrelevant. For example, the widely-used GIST882 line (created in our lab), is MAX-wildtype but has homozygous *RB1* and *TP53* intragenic deletions, resulting in complete loss of RB1 and TP53 protein expression. Similarly, our GIST430 line has high-level *CDK4* and *MDM2* genomic amplification, inactivating RB1 and TP53, respectively. These downstream events render MAX experimental manipulations functionally irrelevant, at least with respect to assessing biologic impact of p16 down-regulation.

9) “In the RNA-seq experiments, are signatures related to Myc perturbed? This may help to clarify if the effects of MAX are Myc dependent or independent. This could be addressed by integrating the transcriptional data with Myc chip-seq binding data etc”. To follow up on this, we performed additional analyses of the RNAseq data, to identify whether MYC signatures were altered upon MAX restoration. Gene Set Enrichment Analysis (GSEA-Subramanian et al, 2005) showed that none of the 187 signatures from the Oncogenic Signatures Collection of the Molecular Signatures Database (MSigDB-C6) was significantly upregulated or downregulated at a FDR <5% and a nominal p-value <5%. Furthermore, GSEA using 24 published MYC-related gene expression signatures (from collections C2, C4, C5 and C6 of MSigDB) indicates that none of the signatures was upregulated or downregulated at a FDR <5%. These results indicate that MYC-related gene sets are not enriched amongst the genes differentially expressed in GIST48 cells after MAX restoration, and therefore suggest that MAX tumor suppressor roles, in GIST, are not necessarily MYC-dependent. We have now added a statement to the text, summarizing the negative results of these computational analyses.

We would be delighted to respond to any additional criticisms that might arise in re-review of the manuscript.

REVIEWERS' COMMENTS:

Reviewer #3 (Remarks to the Author):

The authors have done an excellent job in responding to my original concerns.